# Structural Networks for Brain Age Prediction

**Oscar Pina**[1]                                                        OSCAR.PINA@UPC.EDU
**Irene Cumplido-Mayoral**[2]                                    ICUMPLIDO@BARCELONABETA.ORG
**Raffaele Cacciaglia**[2]                                        RCACCIAGLIA@BARCELONABETA.ORG
**José María González-de-Echávarri**[2]                   JGONZALEZ@BARCELONABETA.ORG
**Juan Domingo Gispert**[2]                                    JDGISPERT@BARCELONABETA.ORG
**Verónica Vilaplana**[1]                                        VERONICA.VILAPLANA@UPC.EDU
[1] *Technic University of Catalonia (UPC)*
[2] *Barcelona Brain Research Center (BBRC)*

**Editors:** Under Review for MIDL 2022

## Abstract

Biological networks have gained considerable attention within the Deep Learning community because of the promising framework of Graph Neural Networks (GNN), neural models that operate in complex networks. In the context of neuroimaging, GNNs have successfully been employed for functional MRI processing but their application to ROI-level structural MRI (sMRI) remains mostly unexplored. In this work we analyze the implementation of these geometric models with sMRI by building graphs of ROIs (ROI graphs) using tools from Graph Signal Processing literature and evaluate their performance in a downstream supervised task, age prediction. We first make a qualitative and quantitative comparison of the resulting networks obtained with common graph topology learning strategies. In a second stage, we train GNN-based models for brain age prediction. Since the order of every ROI graph is exactly the same and each vertex is an entity by itself (a ROI), we evaluate whether including ROI information during message-passing or global pooling operations is beneficial and compare the performance of GNNs against a Fully-Connected Neural Network baseline. The results show that ROI-level information is needed during the global pooling operation in order to achieve competitive results. However, no relevant improvement has been detected when it is incorporated during the message passing. These models achieve a MAE of 4.27 in hold-out test data, which is a performance very similar to the baseline, suggesting that the inductive bias included with the obtained graph connectivity is relevant and useful to reduce the dimensionality of the problem.

**Keywords:** Graph Neural Network, Graph Signal Processing, Structural MRI, Brain Age

## 1. Introduction

Geometric Deep Learning (GDL) (Bronstein et al., 2021) attempts to exploit geometric structures to reduce the dimensionality of the learning task. This idea is behind most successful DL architectures such as Convolutional Neural Networks (CNN) which operate on 2D-grids and Recurrent Neural Networks (RNN) for sequences. Geometric priors are split into two principles (Bronstein et al., 2021): (i) symmetries or transformations that preserve certain properties, which depend on the level of structure of the domain and (ii) scale separation or multiscale representation. Another example are Graph Neural Networks (GNN), which operate on graphs and have boosted their popularity in recent years due to their learning potential and flexibility. The symmetry demanded to GNN layers is to be

equivariant to the permutation of the nodes, whereas a global pooling for graph-level tasks must be permutation-invariant (Hamilton).

Neuroimaging data is usually high-dimensional and including a geometric inductive bias is needed to break the curse of dimensionality. Although parcellations are used to reduce the problem from voxel-level to region of interest (ROI)-level, the dimensionality may still be too high, for instance in functional-MRI (fMRI), where not only are ROIs taken into account but also their interactions. There is increasing literature using GNNs for fMRI analysis, such as disease prediction, since the so-called Functional Connectivity Matrices (FCM) can be seen as adjacency matrices (Kim and Ye, 2020) (Li et al., 2021).

Graph Neural Networks have also been applied to structural MRI (sMRI) at different levels: (Besson et al., 2021) used Graph Convolutional Netowrks (GCN) to predict sex and age from the cortical surface area, represented as a mesh, in order to find a relationship between cortical folding and demographic features. (Parisot et al., 2017) performed semi-supervised node prediction by building a population graph whose connectivity was defined via phenotypic similarities between subjects (nodes of the population graph) and the structural profile of each subject as node features. However, to the best of our knowledge, GNNs have not been used at ROI level for sMRI analysis.

Our goal is to create a graph whose nodes are brain ROIs and whose connectivity depends on ROI volume dependencies, which are extracted from sMRI data using tools from Graph Signal Processing (GSP) literature. Formally, the task is the definition of the connectivity of a graph of N nodes, given a data set of N-dimensional data points $D = \{x^i\}$ where $x^i \in \mathbb{R}^N$ so that GNNs can be applied in this new geometric domain to leverage the inductive bias included in the graph topology. Within this construction, each subject represented by a N-dimensional vector of ROI volumes will be seen as a graph signal defined on the N vertices of the obtained network. We also aim to analyze whether these techniques can be used for supervised downstream tasks via GNNs. To do so, we compare most popular methodologies for the construction of the graph as well as different GNN-based models for brain age prediction given the structural profile of a subject. The code is publicly available at https://github.com/imatge-upc/structnet_aging.

## 2. Methods

### 2.1. Graph Topology Learning

The objective is to obtain the connectivity of a graph of N nodes, which correspond to N brain regions, where edges represent statistical relationships between the volumes of those regions. Relationships will vary according to the methodology used to learn the structure and the constraints of the optimization problem. Hence, given the structural information (N ROI brain volumes) of M subjects $D = \{x^i\}$, $\forall i = 1...M$, $x^i \in \mathbb{R}^N$, arranged in a matrix $X_D \in \mathbb{R}^{N \times M}$, the task consists of inferring the underlying graph structure of the data, represented by a $N \times N$ adjacency matrix.

#### 2.1.1. CORRELATION BASED TOPOLOGY

A strategy to obtain a graph structure could be using correlation. The $N \times N$ correlation matrix of the (centered, standardized) sample is obtained via $R_x = X_D X_D^T$. However, it is

known that correlation networks may declare edges due to confounders, that is, two vertices $i, j$ may be connected due to the effect a third one $k$ has on both $i$ and $j$ so extra conditions must be considered (Stanković et al., 2019). One solution is to add sparsity constraints to obtain an optimization problem. If we consider an arbitrary vertex $i$, being $y_i$ the set of observation of this vertex and $\omega_{ij}$ the weight of the edge from $i$ to node $j$, the weights can be obtained by minimizing: (Stanković et al., 2019):

$$J_i = ||y_i - \sum_{j \neq i} \omega_{ij} y_j||_2^2 + \lambda \sum_{j \neq i} |w_{ij}| \tag{1}$$

where the first term promotes the correlation between connected vertices and the second term, L1 regularization, promotes the sparsity of the graph (Stanković et al., 2019). In order to derive the entire graph structure, this optimization problem must be run for all vertices or features of the data.

### 2.1.2. GRAPHICAL LASSO

A second approach for the construction of the graph structure is Graphical LASSO. The algorithm attempts to estimate the precision matrix $(Q)$, which is the inverse of the covariance matrix $(Q = \Sigma^{-1})$, by minimizing the function:

$$J(Q) = tr\{QR_x\} - log(detQ) + \lambda||Q||_1 \tag{2}$$

The term $tr\{QR_x\}$ quantifies the smoothness of the data in the graph, that is, the sum of the differences between graph signals values of adjacent nodes. Secondly, $log(detQ)$ is added to force the matrix to be symmetric positive definite, so that its eigenvalues must be all positive. Note that $detQ$ is the product of the eigenvalues. Finally, the last term, controlled by the parameter $\lambda$, forces the sparsity of the matrix.

### 2.2. GNNs and Permutation Invariance/Equivariance

Graph Neural Networks (GNN) are a framework for the application of neural networks to graph-structured data. There are various theoretical motivations, but most popular GNN architectures are under the umbrella of the Message Passing framework (MPNN) (Gilmer et al., 2017). In each convolutional layer, nodes send messages to their neighbors, aggregate all the messages received and update their hidden representation. During the message-passing (Gilmer et al., 2017) operation of a GNN layer $l$, the node update equation for an arbitrary vertex $x_i$ can be expressed in terms of the message function $M_l$, aggregation function $F_{j \in N(i)}$ and update function $U_l$ as:

$$x_i^{(l)} = U_l \left( x_i^{(l-1)}, F_{j \in N(i)} \{M_l(x_i^{(l-1)}, x_j^{(l-1)})\} \right) \tag{3}$$

Overall, this operation must be permutation-equivariant. Recall that a function $f$ is permutation-equivariant if given a permutation matrix $P$, $f(PX, PAP^T) = Pf(X, A)$ and $g$ is permutation-invariant if $g(PX, PAP^T) = g(X, A)$. When we are working with sets (with no structure A), $f$ and $g$ are permutation-equivariant and permutation-invariant, respectively, if $f(PX) = Pf(X)$ and $g(PX) = g(X)$. At the same time, the aggregation of the set of messages received by a node $(X_{N(i)})$ must be permutation-invariant:

$F(PX_{N(i)}) = F(X_{N(i)})$. Most common implementations of this aggregation operation are the sum $F_{j\in N(i)} = \sum_{j\in N(i)}$ or the max function $F_{j\in N(i)} = \max_{j\in N(i)}$, for example.

When GNNs are used in a graph-level task, which consists of predicting a value for each input graph, a Global Pooling operation is used to aggregate all node representations of the graph $(X)$ into one single vector, which needs to be permutation-invariant as well: $F(PX) = F(X)$. Then, the entire model which consists of Message-Passing layers followed by the Global Pooling is permutation-invariant.

However, MRI ROI-based graphs may be exempt of this permutation-equivariance rule because we can ensure all graphs will have exactly the same number of nodes and each node is an identifiable entity itself, so that we can establish an arbitrary order and keep the same criterion for all graphs of the data set. To test whether adding ROI information into the message passing process or the global pooling is useful or not, we have compared the performance of four different GNN-based models, corresponding to all combinations of ROI awareness and permutation equivariance/invariance in Message Passing and Global Pooling layers, in the downstream task of brain age prediction. A list of all GNN-based models trained and the Fully-Connected Neural Network (FCNN) baseline can be found in Table 1.

Table 1: GNN based models

| Model | Parameters | Message-Passing | Global Pooling |
|---|---|---|---|
| FCNN | 1729 | - | - |
| GNN *PE-PI* | 1503 | Permutation equivariant | Permutation invariant |
| GNN *PE-RA* | 1711 | Permutation equivariant | ROI-aware |
| GNN *RA-PI* | 1603 | ROI-aware | Permutation invariant |
| GNN *RA-RA* | 1811 | ROI-aware | ROI-aware |

The permutation-invariant Message Passing model used is the well-known GIN model (Xu et al., 2018), whose node-level function at convolutional layer $(l)$ is defined as:

$$x_i^{(l)} = h_\Theta^{(l)}\left((1 + \epsilon^{(l)})x_i^{(l-1)} + \sum_{j\in N(i)} x_j^{(l-1)}\right) \tag{4}$$

where $h_\Theta^{(l)}$ is a neural network parametrized by $\Theta$, implemented as a Multi-Layer Perceptron (MLP), $\epsilon^{(l)}$ is a learnable parameter and $x_j^{(l-1)}$ is the hidden representation of node $j$ after the layer $(l-1)$. Therefore, following the Message Passing formalism, $M_l$ is the identity function, the aggregation operation is the sum $F_{j\in N(i)} = \sum_{j\in N(i)}$ and the update function $U_l$ is implemented by the MLP $h_\Theta$. The permutation invariant Global Pooling operation is the concatenation of the *max* and *sum* functions:

$$g_x = \left[\max_{\forall i \in G} x_i^{(L)}, \sum_{\forall i \in G} x_i^{(L)}\right] \tag{5}$$

The ROI-aware Message Passing layer has been implemented using a variation of the GNN vision of the PointNet++ (Qi et al., 2017) network, proposed for point clouds in

metric spaces with non-uniform distribution. Here, the Message-Passing scheme used at convolutional layer $l$ is:

$$x_i^{(l)} = \gamma_\Theta^{(l)} \left( \sum_{j \in N(i) \cup \{i\}} \phi_\Theta^{(l)}(x_j^{(l-1)}, p_j - p_i) \right) \qquad (6)$$

Where $\gamma_\Theta^{(l)}$ and $\phi_\Theta^{(l)}$ are neural networks parametrized by $\Theta$, implemented as MLPs, $x_j^{(l)}$ is the hidden representation of node $j$ after the layer $(l-1)$ and $p_i$, $p_j$ are one-hot vectors identifying the ROI nodes $i$ and $j$ represent, respectively. The initial purpose of $p_i$ is to define the point position in a metric space and $p_j - p_i$ is the distance between those two points. Since we are not in such a metric space, being $p_i$ one-hot vectors and recalling that the graph connectivity will be the same for all samples, we expect $p_j - p_i$ to encode edge information. Note that the sign will determine the direction since the message from an arbitrary node $i$ to one of its neighbors $j$ will have the term $p_j - p_i$ whereas the message from $j$ to $i$ will input $p_i - p_j$.

Finally, the ROI-aware Global Pooling layer has been implemented using a weighted sum of all node feature vectors, leading to the following equation for the vector representation of a given graph:

$$g_x = \sum_{i \in G} w_i x_i^{(L)} \qquad (7)$$

where $x_i^{(L)}$ is the feature vector of node $i$ after $L$ convolutional layers and $w_i$ are learnable parameters of the model. Moreover, multiple heads of this layer have been concatenated in order to increase the representation power. Another strategy could have been the concatenation of all node representations but it would have increased the dimensionality of the output representation and we are interested in comparing models with similar number of parameters.

## 3. Experiments and Results

The experiments and results obtained are explained in this section. The pipeline for a brain age prediction experiment consists of the following steps, (i) partition the data into training, validation and test sets (3.1), (ii) define the network topology using the training split (3.2), (iii) fit a model for brain age prediction using the training data and keep the checkpoint that performs the best on validation set and (iv) evaluate the model on test set (3.4). We have carried out this procedure for 5 different data split and model initialization random seeds and the results are shown in section 3.4. Models are implemented in PyTorch (Paszke et al., 2019) and PyTorch Geometric (Fey and Lenssen, 2019).

### 3.1. Data

The data used in our experiments is a subset of n=24106 cognitively unimpaired subjects that range from 45 to 82 years old from the UKBioBank database, subjects with mental diseases or trauma diagnosed have been removed. The information available for every subject are demographic features such as sex, age and education years, as well as the APOE status, the major genetic risk for Alzheimer's disease, and their brain structural parcellation

obtained with FreeSurfer (see Appendix A, Table 4). The atlases of the parcellation are the Desikan-Kiliany Atlas (Desikan et al., 2006) for the cortex and the Aseg (Fischl et al., 2002) for the subcortex. We have discarded features that do not represent single ROI volumes, which has led us to N=104 ROI volume features per subject. Moreover, ROI-volumes have first been adjusted by the Total Brain Volume of each subject and standardized before generating the connectivity matrices. In our experiments, we have used 60% of the data for defining the network topology and training the model, 20% for the validation split used during the brain age prediction task and 20% as hold-out test data.

### 3.2. Graph Topology Learning

In this section we provide a qualitative and quantitative comparison of the techniques used to create the graph topology. A study of the effects of the sparsity hyperparameters when defining the structure in terms of multiple network metrics such as node degree, clustering coefficient and shortest path length can be found in Appendix B.

We start with the correlation-based approach with sparsity constraints. The sparsity of the network is controlled by $\lambda$, since it is the parameter multiplying the second term of Equation (1), which promotes sparsity of the weights. Hence, the larger $\lambda$, the more sparse the graph will be.

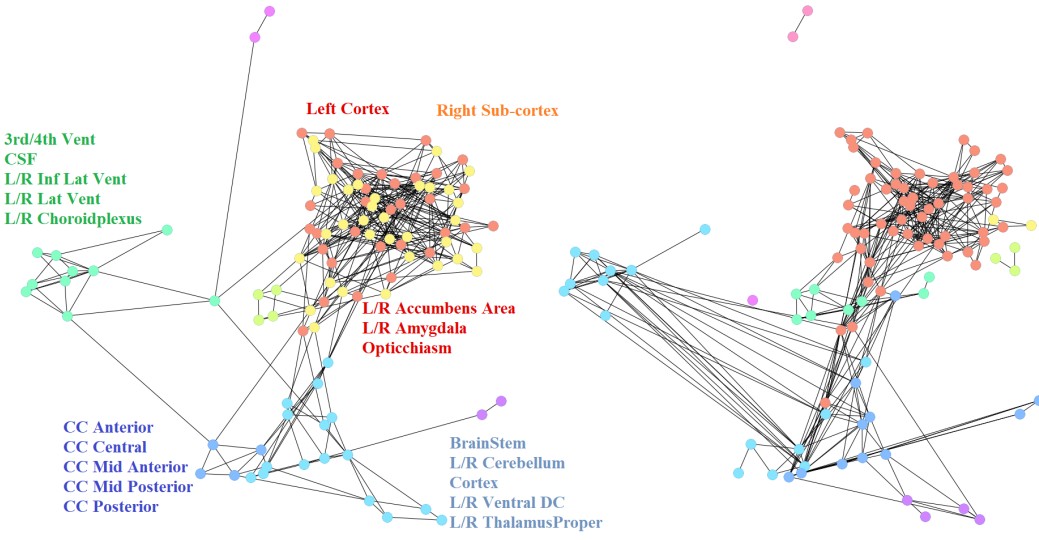

Figure 1: Networks generated with (left) Correlation-based approach ($\lambda = 5 \times 10^{-2}$) and (right) Graphical LASSO ($\lambda = 3 \times 10^{-1}$)

After analyzing the effect of the sparsity hyperparameter to select the optimal value (see Appendix B), the resulting network is shown in Figure 1 (left). For a deeper visual understanding of the network, nodes are labeled according to a label propagation community detection algorithm (Cordasco and Gargano, 2011), which initially assigns a color to each node so that no two adjacent vertices share the same label and iteratively updates node

labels according to the most frequent labels of their neighbors until convergence. Cortical and subcortical regions build two sub-networks with few connections between them, being the accumbens area and the amygdala the ROIs connecting both of them. Moreover, within the subcortex we can also observe multiple independent communities. For example, all corpus callosum ROIs are clustered together and another cluster consists of the ventricles. On the other hand, the cortex sub-network is much more dense. It turns out that each left-side cortical ROI is connected to its right-side counterpart. However, the community detection algorithm splits the cortex into left-cortex and right-cortex because a part from its counterpart, spatially adjacent ROIs are connected.

Table 2: Volume and Cut Size metrics between Cortical and Subcortical sub-networks

| Method | $\lambda$ | Size | Cortex Vol | Subcortex Vol | Cut-size |
|---|---|---|---|---|---|
| Correlation-based | $5 \times 10^{-2}$ | 378 | 587 | 169 | 18 |
| Graphical LASSO | $3 \times 10^{-1}$ | 335 | 426 | 244 | 13 |

The other strategy used has been Graphical LASSO, which also includes a sparsity parameter $\lambda$. This method is more sensitive to $\lambda$ when working with sMRI compared to the previous approach. The obtained network is shown in Figure 1 (right), which has been plotted employing the same layout as Figure 1 (left) for visual purposes. There are slight differences between the networks; it can be observed that although the connections between cortical and subcortical regions are similar, the density of ROIs of the subcortex has increased with respect to the previous approach at the expense of a decrease in the number of connections between cortical ROIs. These observations can also be quantitatively obtained from Table 2, in which *Method* and $\lambda$ denote the strategy and the value of the sparsity parameter employed, *Size* is the number of edges of the network, *Cortex Vol* and *Subcortex Vol* are the volume, or sum of the degrees, of the cortical and subcortical ROIs, respectively. Finally, *Cut-size* is the number of edges connecting both sub-networks. We can observe that the cut-sizes between cortex and subcortex of the networks using both methods are similar, but there are differences between cortex and subcortex volumes.

### 3.3. Group Structural Differences

The definition of structural networks opens a new line of research for the study of structural differences between groups of subjects, such as APOE-$\epsilon$4 carriers vs non-carriers. It can be achieved by learning a topology for each group and employing network (dis-)similarity metrics to quantify differences between these networks. A more detailed description of the approach can be found in Appendix C. No structural differences have been found between the networks of the groups. Based on previous studies using structural source-based morphometry and independent component analysis (Cacciaglia et al., 2020), we have further looked into left and right cortical sub-networks and we have observed an increase in the volume of the right-side for APOE-$\epsilon$4 carriers with respect to non-carriers right-cortex, which is in line with results obtained in (Cacciaglia et al., 2020). Finally, in ROI-graph setting we can analyze ROI by ROI whether networks are similar or not and we have observed that node centrality of the left-insula and the right-parahippocampal are higher in networks generated with APOE non-carrier subjects.

### 3.4. Brain Age Prediction

Table 3 summarizes the Brain Age prediction performance of all GNN based models following the Correlation-based and Graphical LASSO topology learning strategies as well as the baseline. The models are evaluated on hold-out test data and Table 3 entries are the mean and standard deviation of the results for 5 different data split and model initialization random seeds. The training set is used for both (i) defining the network topology and (ii) training the model, and the validation set is employed to determine the best checkpoint during training. The baseline consists of a 1 hidden layer FCNN and all models have been trained using Adam optimizer with a learning rate of $\alpha = 1 \times 10^{-3}$, a dropout with probability $p = 0.2$ has been included. Performance when following any of the topology creation approaches is very similar, which suggests that the strategies similarly capture the dependencies of the data. Results show that a ROI-aware global pooling operation is crucial in order to achieve competitive results in terms of MAE and Pearson Correlation ($\rho$) between predictions and true age ($MAE = 4.3$ years and $\rho = 0.7$ with ROI-aware global pooling vs $MAE = 6$ years and $\rho = 0.3$ with permutation-invariant global pooling). Therefore, ROI information is required when pooling nodes to one single vector representations. However, ROI information seems to have no relevance during the message passing operation since no variation has been detected when it is included or not. Models implementing a ROI-aware Global Pooling slightly improve FCNN Baseline performance so the ROI-volume interactions captured by the graph are sufficient to achieve competitive results.

Table 3: Brain Age Prediction Results

| Model | Method | MAE | Pearson Correlation |
|---|---|---|---|
| FCNN | - | 4.39 ±0.04 | 0.70 ±0.01 |
| GNN *PE-PI* | Graphical LASSO | 6.01 ±0.09 | 0.29 ±0.04 |
| GNN *PE-PI* | Correlation-based | 5.99 ±0.06 | 0.30 ±0.03 |
| GNN *PE-RA* | Graphical LASSO | 4.31 ±0.11 | 0.69 ±0.03 |
| GNN *PE-RA* | Correlation-based | 4.27 ±0.07 | 0.70 ±0.01 |
| GNN *RA-PI* | Graphical LASSO | 6.08 ±0.12 | 0.26 ±0.05 |
| GNN *RA-PI* | Correlation-based | 5.99 ±0.09 | 0.30 ±0.04 |
| GNN *RA-RA* | Graphical LASSO | 4.34 ±0.04 | 0.69 ±0.01 |
| GNN *RA-RA* | Correlation-based | 4.36 ±0.06 | 0.69 ±0.01 |

### 4. Conclusions

In this manuscript we have followed a graph-based approach for the analysis of brain structural networks by implementing techniques from GSP for the learning of a graph topology. GNNs have been able to leverage the inductive bias of the networks and slightly improve the performance of a FCNN with the same number of parameters on the task of brain age prediction. We have also seen that permutation invariance is not a desirable property for a Global Pooling layer when working with ROI-graphs. These results encourage the usage of sparse approaches against dense methods, the definition of more powerful GNN layers is a trending line of research and more expressive power could make that extra difference.

## Acknowledgments

This work has been supported by the Spanish Research Agency (AEI) under project PID2020-116907RB-I00 of the call MCIN/ AEI /10.13039/501100011033 and the FI-AGAUR grant funded by Direcció General de Recerca (DGR) of Departament de Recerca i Universitats (REU) of the Generalitat de Catalunya.

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

## Appendix A. Demographic Data

Table 4: Sample Demographics

| Subjects N | Age mean (std) | Sex, female N (%) | Education Years mean (std) | APOE-$\epsilon 4$ carriers N (%) |
|---|---|---|---|---|
| 24106 | 64.28 (7.50) | 12406 (51.64) | 17.15 (2.64) | 6755 (28.02) |

## Appendix B. Sparsity Hyperparameters

In this section we study the effect of the sparsity hyperparameter $\lambda$ when learning a graph topology and explain the criterion we have followed to select an optimum value, independently for each strategy. We compare density and other network metrics for a range of values of $\lambda$ in order to find the setting that leads to the most desirable network. Small-world networks are defined as high clustered networks but with small shortest path lenghts (Watts and Strogatz, 2011). We are interested in a connected, sparse, small-world network.

The density of an undirected graph is defined as $d = \frac{2E}{N(N-1)}$ where $N$ is the number of nodes and $E$ the number of edges. Note that the density ranges from 0, when there are no connections, to 1 if the graph is complete. Low density networks will only consider the strongest interactions of the data whereas high densities would include too many interactions so that the dimensionality of the problem would not be reduced. Figure 2 shows how the density of the obtained graphs varies according to $\lambda$. The lower $\lambda$, the more dense the networks are. However, Correlation-based networks converge to a density lower than 0.2 whereas the output of Graphical LASSO ends up being the complete graph (density 1) when $\lambda$ is too small. It can also be observed in Figure 3, where node degree distributions converge to $N - 1$, meaning that the graph is complete.

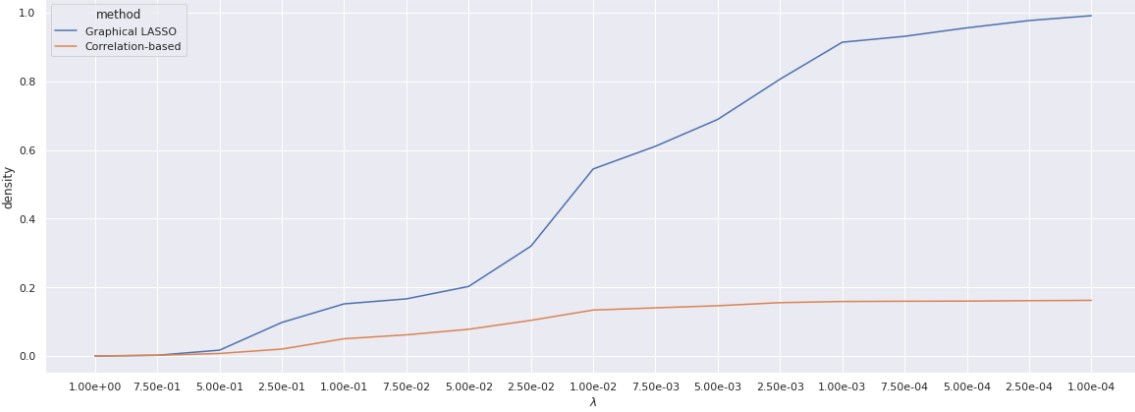

Figure 2: Graph density as function of $\lambda$

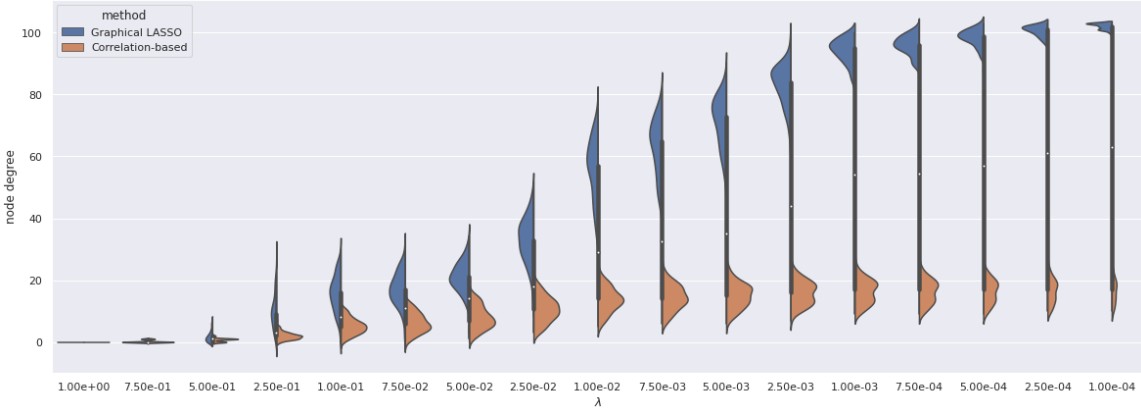

Figure 3: Graph node degree distributions as function of $\lambda$

The shortest path length between two nodes $i$ and $j$ measures the minimum number steps needed to reach $j$ starting from $i$. For a given node $i$, one can compute the average of the shortest path lengths starting from $i$ to all other vertices and get the distribution for all nodes. For disconnected graphs with more than one connected component, we have set the distance between two non-reachable nodes to be $N$, the number of nodes. Figure 4 shows that for Graphical LASSO, between $\lambda = 2.5e - 1$ and $\lambda = 5e - 1$ the graph rapidly passes from being a purely disconnected network (average short path length near $N$) to being an almost complete network (average short path length near 1), so we may be interested in a $\lambda$ between those values. The Correlation-based method has a similar behavior but not as abrupt and for smaller values of $\lambda$, between $5e - 2$ and $2.5e - 1$.

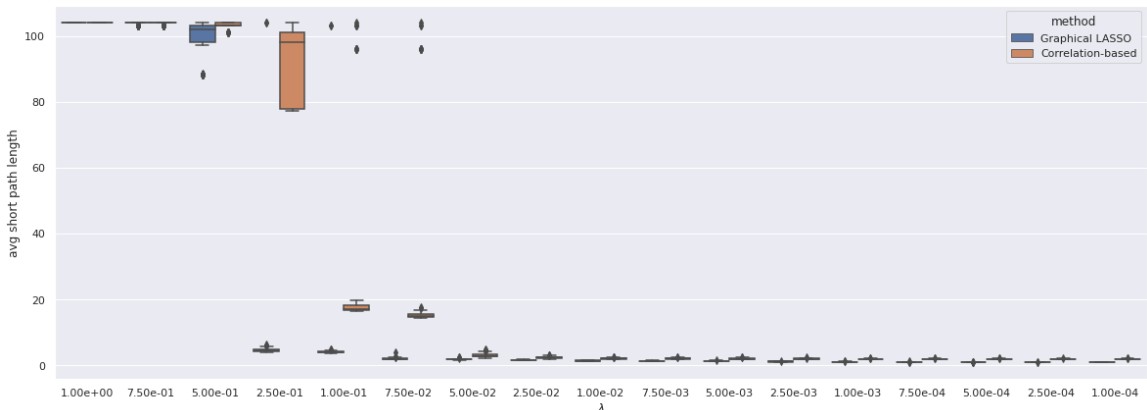

Figure 4: Graph shortest path length distributions as function of $\lambda$

Clustering coefficient quantifies how nodes tend to cluster together. It can be computed globally (also known as transitivity) as $C = 3 \times \frac{T(G)}{num\ of\ triplets}$ or locally, for each node $i$ as

$C_i = \frac{T(i)}{deg(i) \times (deg(i)-1)}$ where $T(i)$ is the number of triangles $i$ takes part and $deg(i)$ is the degree of the node. Once again, we can compute this value locally and compare distributions as shown in Figure 6. Recall that we are interested in high clustering coefficient values, but without excessively compromising the sparsity of the network, which leads us to deduce that $\lambda \in (1e-1, 5e-1)$ for Graphical LASSO and $\lambda \in (2.5e-2, 1e-1)$ for Correlation-based method seem reasonable values.

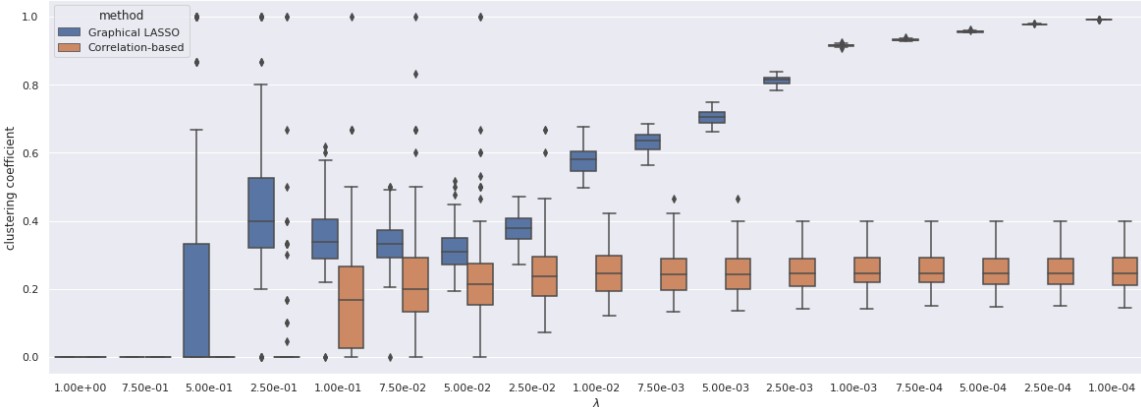

Figure 5: Graph clustering coefficient distributions as function of $\lambda$

Betweenness centrality is a node-level metric defined as the ratio of shortest paths of the graph that pass through a vertex. Nodes that show a high centrality may be in the intersection between multiple communities, for example. We expect node centralities to be low for the majority of the vertices, but to have a few nodes with higher values, identified as hubs. These hubs are the outliers of the boxplots of Figure 6 when $\lambda \in (2.5e-2, 2.5e-1)$.

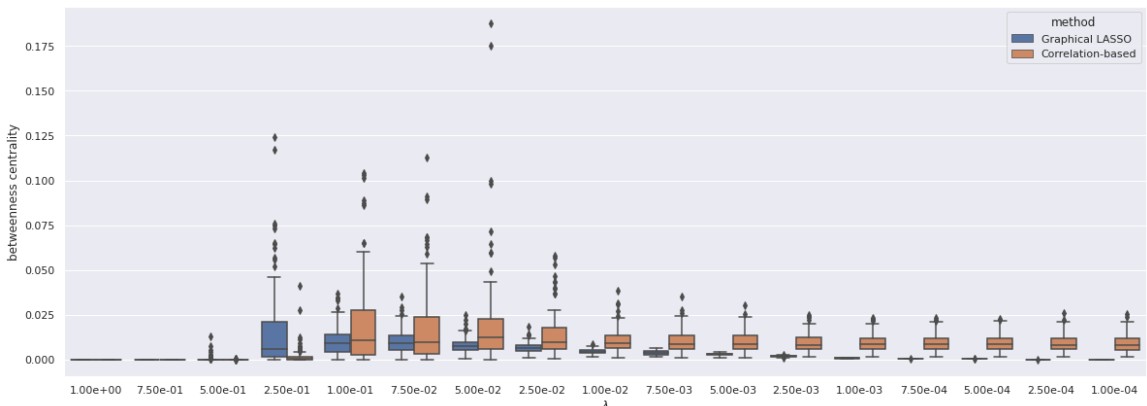

Figure 6: Graph betweenness centrality distributions as function of $\lambda$

To conclude our study, in order to select a value for $\lambda$ constrained to the ranges that offer the desirable structural properties, we have selected the value of $\lambda$ that provides a better

$\sigma$, which compares the average shortest path length and the average clustering coefficient of the network ($L$ and $C$, respectively) vs these metrics obtained from a random graph ($L_r$ and $C_r$, respectively) $\sigma = \frac{C}{C_r} \frac{L_r}{L}$. The optimum values have been $\lambda = 3e-1$ for Graphical LASSO and $\lambda = 5e-2$ for the Correlation-based method.

## Appendix C. Group Structural Differences

In this appendix we explore in more detail the approach of detecting variations in the structural connectivity matrices generated with subjects from different groups (Section 3.3). To do so, we analyze dissimilarities between APOE-$\epsilon$4 carriers and non-carriers structural networks obtained following the Correlation-based methodology with an sparsity hyperparameter of $\lambda = 5e-2$. Since networks are only generated using 60% of the data, we have run this experiment multiple times with different splits. Moreover, the groups are imbalanced so for each split we down-sample the most prevalent class so that the number of subjects is equal for both groups. Figure 7 shows the obtained networks which have been plotted using the same template as Figure 1 and running the same community detection algorithm.

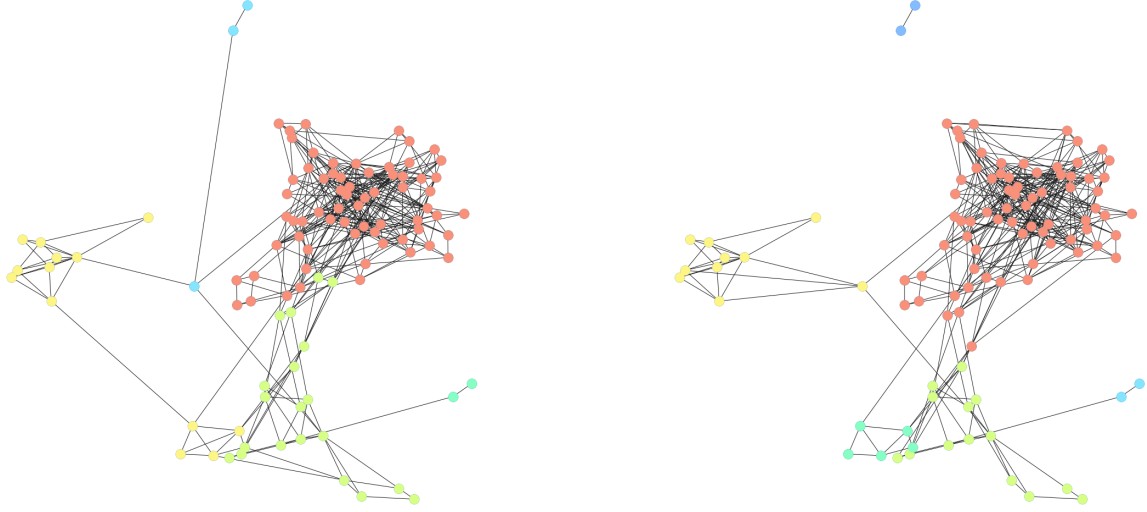

Figure 7: APOE-$\epsilon$4 Non-Carriers (left) and APOE-$\epsilon$4 Carriers (right) Correlation-based Networks

The problem of network dissimilarities can be quantitatively addressed by comparing node-level metric distributions, such as node degree and node centrality (Schieber et al., 2017). In order to quantify the differences between the probability distributions we have made use of the Jensen–Shannon divergence (JS-divergence). The average JS-divergence between node degree distributions of carriers vs non-carriers for networks generated with different splits is 0.03. When comparing node centrality distributions, the averaged divergence is even smaller, 0.004. Hence, we can conclude that both connectivities are very similar.

In a ROI-graph we can split nodes into cortical and subcortical ROIs. Figure 8 shows the histogram of subcortex (left) and cortex (right) volumes for APOE carries vs non-carriers created networks for multiple partitions of the data. It can be observed that major discrepancies are in the cortex area, which is more densely connected (higher volume) in networks generated with APOE carriers. This is at the expense of a slightly lower volume in subcortex regions. At the same time, one can further divide the cortex as left-side and right-side and observe that the cortical differences are not uniform across the entire cortex but they are caused by a more connected right-cortex sub-network for APOE carriers vs non-carriers, as shown in Figure 9.

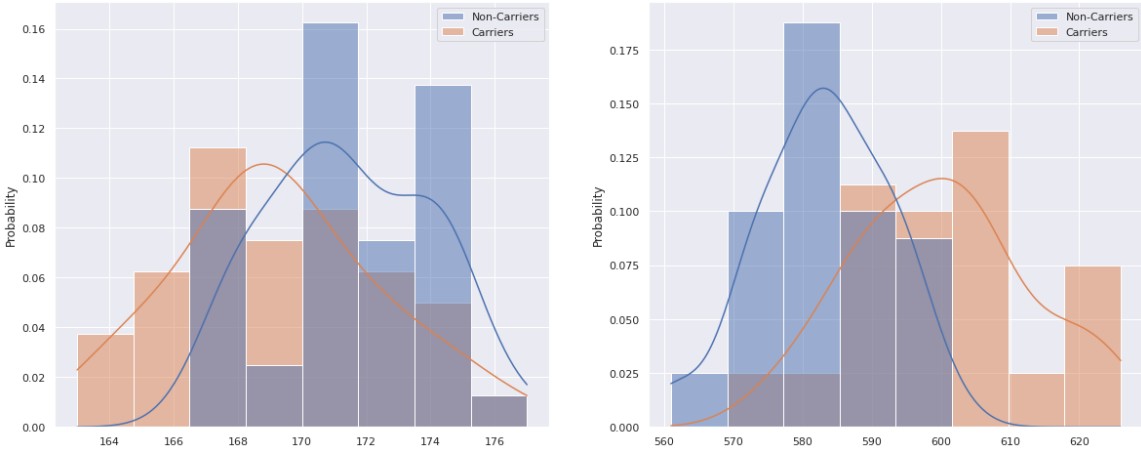

Figure 8: Subcortex (left) and Cortex (right) network volumes for APOE carriers vs non-carriers

Finally, despite the fact that no divergences have been found in either node degree nor node centrality distributions between groups, it is also possible to analyze ROI by ROI whether these metrics differ between groups. We have observed that node centrality of the left-insula and the right-parahippocampal are higher in networks generated with APOE non-carrier subjects. Figure 10 and Figure 11 show the boxplot of the node centrality and node degree, respectively, for some cortical and subcortical ROIs. It can be observed that the left insula and the right parahippocampal present lower centrality in networks created with APOE-non carriers. Variations can also be found on the centrality of subcortical ROIs such as the amygdalas and the CC posterior.

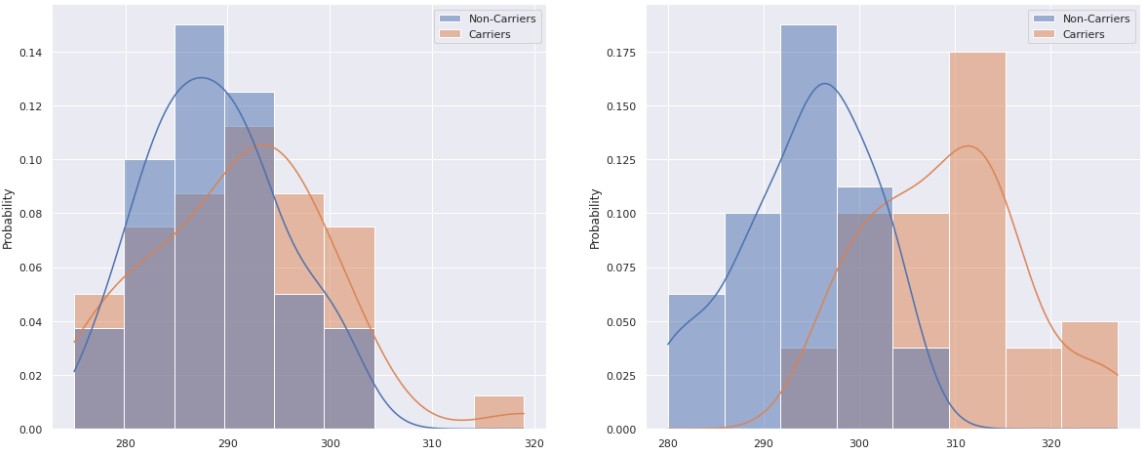

Figure 9: Left-Cortex (left) and Right-Cortex (right) network volumes for APOE carriers vs non-carriers

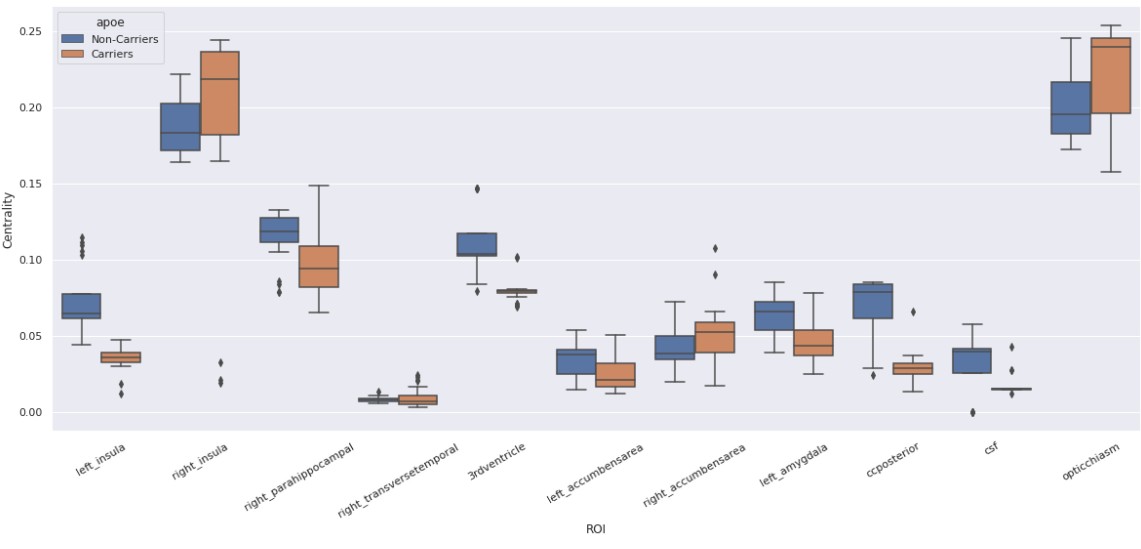

Figure 10: Node Centrality Boxplot APOE Carriers vs Non-carriers

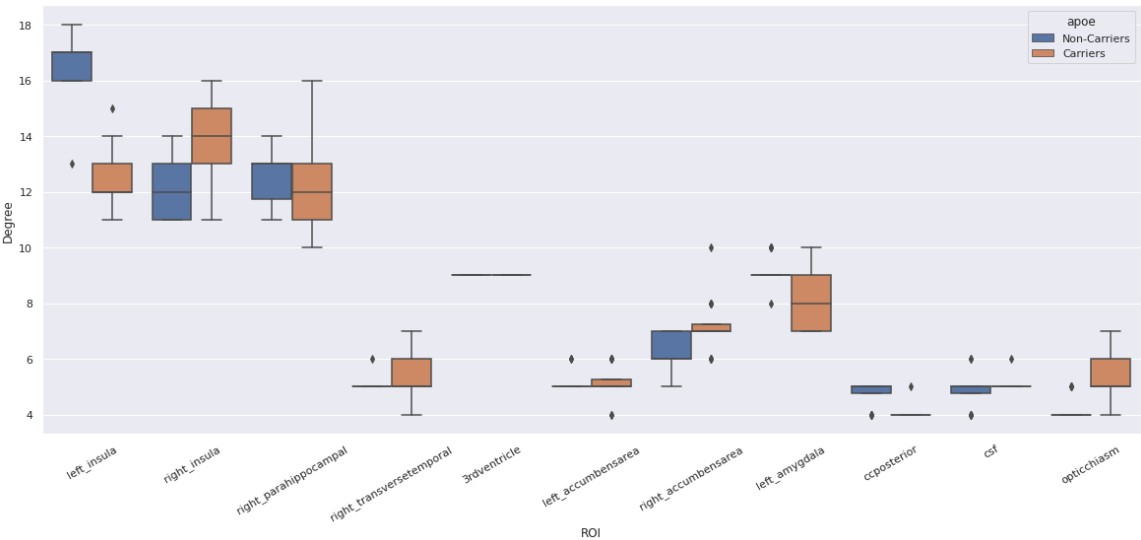

Figure 11:  Node Degree Boxplot APOE Carriers vs Non-carriers

