# OpenReview forum: "Structural Networks for Brain Age Prediction"
_MIDL.io/2022/Conference — MIDL 2022_

### Official Review · Reviewer_dUNQ · 2022-01-14

**Confidence:** 4
**Preliminary Rating:** 3
**Recommendation:** Poster

**Summary:**

The paper represents a graph-based approach for the analysis of brain structural networks by implementing techniques from GSP for the learning of graph topology. Then, a qualitative and quantitative comparison of the resulting networks was obtained with common graph topology learning strategies and finally, a GNN-based model was trained for brain age prediction.

**Strengths:**

1) Increasing interest in the topic of GNN and relevant state-of-the-art work neuroimaging field.
2) Mathematical definitions are well articulated and well linked in the paper.
3) The experimental section (including the appendix) shows the good performance of the proposed method.

**Weaknesses:**

As follows:

1) a few English mistakes and referencing (Figure ??) should be removed before submission.
2) the authors claimed that "There is increasing literature using GNNs for fMRI analysis". I suggest first defining well the analysis of fMRI and second citing some papers that used GNN for this task and explaining the methods proposed. Here it is a bit confusing to analyze an image using a GNN while it is euclidean structured data.
3) in the introduction part it is mentioned: "rarely have GNNs been used at ROI level for sMRI analysis.". I suggest citing existing papers related to this task and performing a comparison between them and the proposed method in this manuscript.
4)   The experimental analysis is not convincing for the dataset, evaluation metrics and comparing algorithms are not enough. More description of the universality of the proposed method should be proved using more experiments.
5) The motivation in the introduction section is not well defined.
6) I cannot see a technical novelty. The authors should clarify the importance of such analysis and why GNN is a good choice for this.

**Deanonymize Review:**

yes

**Final Rating After The Rebuttal:**

4: Weak Accept

**Justification Of The Final Rating:**

I cannot see any novelty in the paper and the authors did not cover all literature related to their problem. However, the writing and the clear answers provided by the authors during the rebuttal convinced me at some point. Therefore, I changed my rating to "Weak Accept".

**Paper Type:**

validation/application paper

**Questions To Address In The Rebuttal:**

1) What is the impact of such analysis in the real world, what is the advantage of leveraging GNN in this case? There is a lack of clarity of the motivation.
2) what is the novelty of this work?
3) the advantage and limitations of this analysis needs to be discussed.

**Special Issue:**

no

---

### Official Review · Reviewer_HbpK · 2022-01-24

**Confidence:** 3
**Preliminary Rating:** 2

**Summary:**

The paper performs a set of experiments comparing different methods of creating brain connectivity graphs from structural MRI data (correlation/graphical LASSO) to different types of graph convolution (GIN, PointNet++) and graph pooling. The task used to make the comparison is brain age prediction (regression) on UKBioBank data.

**Strengths:**

The paper attempts to compare and contrast different methods of constructing brain graphs and how effective each one is when used in different types of graph neural network (GNN) architecture. This type of study serves as a useful benchmark for the community especially when trying to develop new methods using the same type of data and/or task.

**Weaknesses:**

Unfortunately the paper is hard to follow and the experiments section is not clear. The authors claim that the papers main objective is to test how different graph construction methods perform in supervised downstream tasks using GNNs. However, in Section 3.2 label propagation community detection is introduced. Only one baseline is used and it is not explained.

**Deanonymize Review:**

no

**Detailed Comments:**

- In Section 2.1.1 the definition of correlation is only valid if the data has been mean centred which has not been mentioned anywhere previously.


**Final Rating After The Rebuttal:**

2: Weak Reject

**Justification Of The Final Rating:**

I motivation of the paper is still not clear and implementation details are minimal. I think with clearer motivation and more experiments this paper could be important for the community in a future publication.

**Paper Type:**

validation/application paper

**Questions To Address In The Rebuttal:**

- Why does permutation-equivariance matter for brain connectivity matrices since all the nodes are fixed and have the same arbitrary order as specified by a by a pre-defined parcellation? This seems like a redundant experiment.

- What are the ROI features you use to make the structural connectivity matrices? Table in appendix?

- What is the baselines FCNN? What architecture is it? How do you train the network using the connectivity matrices? Thresholded? Lower triangle?

- Where is the explanation of your experiments? How was the data split? What optimisers did you use, and so on?

**Special Issue:**

no

---

### Official Review · Reviewer_Hvfx · 2022-01-26

**Confidence:** 3
**Preliminary Rating:** 3
**Recommendation:** Poster

**Summary:**

The authors evaluated the influence of adding an inductive bias based on the connections between ROIs in a GCN compared to a fully FCNN without connectivity information. To evaluate this, they predictied age using brain structural MRI data.
They used 2 different methods to generate the adjacency matrix that connected the different ROIs, and assessed if the inductive bias introduced in the GCN had influence at two levels of the GCN: message passing and global pooling.
- They found a mild non-significant improvement in the age prediction taks when using the global pooling compared to the FCNN baseline.
- Results are much worse when permutation invariant global pooling is used
- ROI-aware message passing seems irrelevant compared to permuation equivariant

**Strengths:**


- The application of GCNs in brain parcellations is interesting, the results can be useful as a baseline for future network architectures. In particular, the fact that the authors show that "permutation invariance is not a desirable property for a global pooling layer when working with ROI-graphs".

- The experiments were performed on real data from UK Biobank.

- Intereseting ROI-aware implementation of message passing.

- Detailed explanation of the metrics and graph-based methods.

**Weaknesses:**

- I think the papers falls a little short in the methods comparison. Specially, I think adding a more recent tools to generare similarity matrices like similarity network fusion (https://github.com/rmarkello/) would have been interesting.
- Despite the detailed description of the graph-based metrics, the authors do not explain how the training strategy was designed.
- The authors do not detail how the demographic information (sex, year of education...) is used.

**Deanonymize Review:**

no

**Detailed Comments:**

There are some minor errors:
- In Section 3.2 "Figure ?? (right)"
- In Section 3.3, the whole section describes results/draws conclusions without referring to the figures/tables that show them, for example:
"Divergences between node degree and node centrality distributions of both networks are near zero, suggesting that there are no substantial structural differences between them", which figure/table shows this?
and some sentences are not completely clear.


**Final Rating After The Rebuttal:**

4: Weak Accept

**Justification Of The Final Rating:**

The authors have clarified my doubts regarding the generation of the connectivity matrix as well as the training strategy. Which were shared concerns with another reviewer. Moreover, they have clarified section 3.3 in an extended appendix. Despite not having added additional experiments, I think the presented results are interesting enough to be part of MIDL.

**Paper Type:**

validation/application paper

**Questions To Address In The Rebuttal:**

- Have more sophisticaed methods been used to obtain the similarity metrics? Would you be able to incorporate these additional benchmark analysis on the paper?
- What information is used to generate the connectivity matrices? Is the demographic information used for it?
- What was your training strategy? did you use cross-validation? which percentage of the data was used for training?


**Special Issue:**

no

---

### Meta-Review · Area_Chair_6qiy · 2022-02-19

**Recommendation:** Accept (Poster)
**Confidence:** 4

**Metareview:**

The paper examines the impact of graph construction, message passing, and global pooling options in graph neural networks applied to an age prediction task based on automated atlas-based brain MR parcellation. In particular, regularised covariance and graphical lasso are compared, and the impact of permutation equivariance/invariance (in an atlas-based context) is examined. Performance is compared to a baseline fully-connected network.

The work is performed correctly, and the pipeline description and evaluation set-up (data split) has been improved compared to the previous version of the paper. While there is no technical novelty per se, it is valuable to see that no version of the GNN outperforms the baseline, that GNN performance is largely the same for both graph construction approaches, and that global pooling is sensitive to ROI order in the graph.

Pros
* Very large-scale data used (UKB)
* Experiments are informative for other types of graph data whose vertices are in a fixed order

Cons
* Motivation for the choice of baseline, as well as description, is insufficient
* Unclear from the paper - the approach to graph construction from structural imaging is probably suboptimal compared to using multiple morphometric features (see e.g. Seidlitz et al Neuro 2018)

---

### Decision · Program_Chairs · 2022-02-28

Accept